# Adaptable graphitic C$_6$N$_6$-based copper single-atom catalyst for intelligent biosensing

Qing Hong[1,3], Hong Yang[1,3], Yanfeng Fang[1], Wang Li[1], Caixia Zhu[1], Zhuang Wang[1], Sicheng Liang[1], Xuwen Cao[1], Zhixin Zhou[1], Yanfei Shen ●[2] ✉, Songqin Liu[1] & Yuanjian Zhang ●[1] ✉

Self-adaptability is highly envisioned for artificial devices such as robots with chemical noses. For this goal, seeking catalysts with multiple and modulable reaction pathways is promising but generally hampered by inconsistent reaction conditions and negative internal interferences. Herein, we report an adaptable graphitic C$_6$N$_6$-based copper single-atom catalyst. It drives the basic oxidation of peroxidase substrates by a bound copper-oxo pathway, and undertakes a second gain reaction triggered by light via a free hydroxyl radical pathway. Such multiformity of reactive oxygen-related intermediates for the same oxidation reaction makes the reaction conditions capable to be the same. Moreover, the unique topological structure of Cu$_{SA}$C$_6$N$_6$ along with the specialized donor-π-acceptor linker promotes intramolecular charge separation and migration, thus inhibiting negative interferences of the above two reaction pathways. As a result, a sound basic activity and a superb gain of up to 3.6 times under household lights are observed, superior to that of the controls, including peroxidase-like catalysts, photocatalysts, or their mixtures. Cu$_{SA}$C$_6$N$_6$ is further applied to a glucose biosensor, which can intelligently switch sensitivity and linear detection range in vitro.

The establishment of indispensable adaptability plays a crucial role in living systems under a wide variety of environmental stimuli. For example, under normal conditions, thyroid hormones are primarily responsible for the regulation of metabolism at the basal level in humans. Nonetheless, when suddenly exposed to severe cold stimuli, the sympathetic nervous system releases high concentrations of norepinephrine. It significantly accelerates metabolism as the second pathway, which produces more heat to compensate for the cold. It features primary transduction in the normal state and a large gain in exceptional circumstances. For the same reason, mimicking such self-adaptability is highly envisioned in artificial devices[1–3], such as robots, brain-machine interfaces, and the Internet of Things, to freely perceive weak and strong external signals, but remains a grand challenge. From

a chemical point of view, to realize these imperative basic activities and prominent gain effect, seeking catalysts with multiple and modulable reaction pathways is the crux of the matter[4–9]. However, the negative internal interferences and inconsistency of reaction conditions, such as temperature, pH, and solvents, generally exist in different reaction pathways, hampering the effective coordination of multiple catalytic pathways simultaneously[10].

As a metal-free semiconductor, polymeric carbon nitride (PCN) has been intensively explored not only as a metal-free photocatalyst[11–20] but also as a solid ligand to anchor single metal atom[21–23], owing to its engineerable conjugated repetitive units, rich lone pair electrons in the framework, and high physicochemical stability. For example, carbon nitrides with different topological

[1]Jiangsu Engineering Laboratory of Smart Carbon-Rich Materials and Device, Jiangsu Province Hi-Tech Key Laboratory for Bio-Medical Research, School of Chemistry and Chemical Engineering, Southeast University, Nanjing 211189, China. [2]Medical School, Southeast University, Nanjing 210009, China. [3]These authors contributed equally: Qing Hong, Hong Yang. ✉e-mail: Yanfei.Shen@seu.edu.cn; Yuanjian.Zhang@seu.edu.cn

structures (e.g., $C_3N_4$, $C_3N_2$, $C_5N_2$, and $C_2N$)[24–30] and metal dopants (e.g., K and Cu)[31–33] have been developed for a wide range of photocatalytic oxidation reactions[34–37] (e.g., clean water and sanitation) and oxidase[38–41]/peroxidase-like[42–44] activities. Interestingly, these oxidation reactions are catalyzed by carbon nitrides via different reactive oxygen-related intermediates under similar conditions, except for light irradiation[45–49]. As such, we reason that engineered multiformity of carbon nitrides in oxidation processes would offer an intriguing way to solve the inconsistency of reaction conditions and negative internal interference for multiple reaction pathways in realizing self-adaptability; however, to our knowledge, this has rarely been reported.

Herein, we report an adaptable copper single-atom catalyst supported on $C_6N_6$ with a specialized donor-π-acceptor linker ($Cu_{SA}C_6N_6$). $Cu_{SA}C_6N_6$ could not only drive basic oxidation of peroxidase substrates through bound high-valent copper-oxo pathway, but also initiate a second gain reaction under light irradiation via free hydroxyl radical pathway under the same conditions. Moreover, the unique topological structure of $Cu_{SA}C_6N_6$, along with the specified donor-π-acceptor linker, promoted intramolecular charge separation and migration, thus successfully inhibiting the negative interference of electron transfers between the above two pathways. As a result, a sound basic activity and a superb gain up to 3.6 times under household light were obtained, significantly higher than that of its control systems. $Cu_{SA}C_6N_6$ was further successfully applied to a single glucose sensor with intelligent switching of sensitivity and linear detection range in vitro.

## Results

### Synthesis and molecular structure of $Cu_{SA}C_6N_6$

As illustrated in Fig. 1a, copper acetate, as the Cu source, was first complexed with dicyandiamide (DCDA) in ethylene glycol (EG) at 60 °C for 3 h to produce a reddish-brown DCDA-Cu complex. Subsequently, microwave-assisted condensation[36] using EG as the solvent was utilized to synthesize the pale-yellow intermediate, denoted as Cu-$CN_{int.}$, with a yield of 85%. Owing to the pre-coordinated Cu-DCDA complex in EG, the possible formation of a metal or metallic oxide in Cu-$CN_{int.}$ was avoided. As displayed in Supplementary Fig. 1, the X-ray diffraction (XRD) pattern of Cu-$CN_{int.}$ showed the only peak at 26.7°, assigning to interlayer stacking. In contrast, the microwave-assisted polymerization of the blue $Cu^{2+}$ and DCDA mixture resulted in a product with additional XRD peaks for CuO, indicating the significance of pre-complexation in the preservation of the single-atom state. The final product, $Cu_{SA}C_6N_6$, was obtained via thermal polymerization at 550 °C. As a control, $CN_{mw}$ was fabricated by the same microwave-assisted polymerization of DCDA in EG without copper and subsequent thermal condensation.

The combustion elemental analysis in Supplementary Table 1 showed the transformation of the molar C/N value from Cu-$CN_{int.}$ of 0.73 to $Cu_{SA}C_6N_6$ of 0.95, which is ~1. FTIR spectroscopy was used to determine the chemical structure of the new carbon nitride (Fig. 1b). The FTIR spectrum of $Cu_{SA}C_6N_6$ showed strong vibration peaks around 800 and 1200–1700 cm$^{-1}$, assigning to typical triazine rings (CN heterocycles). A characteristic peak at approximately 2900 cm$^{-1}$, which was ascribed to C−H stretching[50], was also observed. Solid-state NMR spectroscopy provided further insights into the nature of the building blocks of $Cu_{SA}C_6N_6$. As shown in Fig. 1c, the carbon atoms in the triazine units corresponded to the peak at ~163 (1) ppm, confirming the existence of conjugated triazine rings in $Cu_{SA}C_6N_6$. The adjacent chemical shift at ca. 156 (2) ppm was assigned to the carbon atom indirectly connected to the triazine ring[51]. Those two types of carbon atoms and FTIR spectrum demonstrated the existence of triazine rings and non-cyclization groups of -N=CH-. The $^1$H magic angle spinning (MAS) NMR spectrum was also employed to determine the local environment of the H atom. As shown in Fig. 1d, the $^1$H NMR spectrum of $Cu_{SA}C_6N_6$ exhibited a main characteristic peak around 8.3 ppm, ascribing to the

-N=CH- group. Notably, this chemical shift of the $^1$H NMR spectrum was often assigned to the aromatic carbon environment[52], indicating the existence of a triazine ring and a second conjugated carbon atom.

The high-resolution X-ray photoelectron spectroscopy (XPS) in Fig. 1e provided additional bonding information. The C 1$s$ XPS spectra exhibited two main peaks at 284.6 eV (C1) and 286.8 eV (C2), which were attributed to the C-C peak and C species in triazine rings (C−N=C), respectively[53]. Interestingly, the carbon peak (ca. 284.6 eV) in PCN and $CN_{mw}$ was assigned to random adventitious carbon, while C-(N)$_3$ shifted to high binding energy at 288.1 eV (Supplementary Fig. 2). These results demonstrated that the non-cyclization group of -N=CH- was coupled, forming -N=CH-CH=N- moieties in the thermal polymerization process. The crystalline texture was further explored using XRD spectroscopy (Fig. 1f). Compared to PCN and $CN_{mw}$, the diffraction peak of $Cu_{SA}C_6N_6$ at ca. 26.0° (002) corresponding to interlayer stacking reflection (002), was broadened and down-shifted, indicating a slightly enlarged interlayer spacing which may arise from the insertion of Cu atoms between $C_6N_6$ layers.

To verify the precise molecular structure of $Cu_{SA}C_6N_6$, matrix-free laser desorption/ionization time-of-flight (LDI-TOF) mass spectra were measured. The m/z peaks resulted from the ablation products of the repetitive $C_6N_6$ units in $Cu_{SA}C_6N_6$. Figure 1g illustrates a series of m/z peaks, including [M + H$^+$] of 127.11, assigning to $C_3N_6H_6$ ($M_1$, melamine, calc.: 126), m/z [M + H$^+$] of 139.15 attributable to $C_4N_6H_6$ ($M_2$, calc.: 138), and m/z [M + H$^+$] of 242.15 attributable to $C_9N_9H_7$ ($M_3$, calc.: 241). The other m/z peaks shown in Supplementary Fig. 3 also supported the ablation unit's information. Therefore, these structural explorations demonstrated that $Cu_{SA}C_6N_6$ featured a repetitive basic triazine core and a -N=CH-CH=N- linker. The possible condensation processes and the molecular structure of $Cu_{SA}C_6N_6$ are shown in Fig. 1h.

### Cu single-atom structure of $Cu_{SA}C_6N_6$

The scanning electron microscopy (SEM) images in Supplementary Fig. 4a shows the disordered and porous structure of $Cu_{SA}C_6N_6$, which distinguishes it from the blocky structures of PCN and $CN_{mw}$ (Supplementary Fig. 4b, c). The high-resolution TEM images in Supplementary Fig. 5 revealed the ultrathin nanosheet-like morphology of $Cu_{SA}C_6N_6$ and no obvious Cu/CuO nanoparticles existed on the surface of the $C_6N_6$ framework, which were in agreement with the XRD results. The large-area high-angle annular dark-field scanning transmission electron microscopy (HAADF-STEM) image (Fig. 2a and Supplementary Fig. 6) corroborated the existence of uniformly dispersed single-atom Cu on the $C_6N_6$ matrix, evident by abundant isolated bright spots highlighted by a white circle. The high Cu content in $Cu_{SA}C_6N_6$ was quantitatively measured by inductively coupled plasma mass spectrometry (ICP-MS) as ~2.36 wt%. The corresponding high-resolution STEM-EDS elemental mapping images showed that the C, N, and Cu species were atomically and homogeneously dispersed across the entire $C_6N_6$ nanosheet (Supplementary Fig. 7 and Fig. 8).

To further investigate the chemical state and local coordination environment of Cu species in $Cu_{SA}C_6N_6$, X-ray absorption fine structure (XAFS) measurements of the Cu K-edge were performed. From the normalized Cu K-edge X-ray absorption near-edge structure (XANES) spectra (Fig. 2b), the absorption edge of $Cu_{SA}C_6N_6$ was located between the control samples of Cu foil and CuO, indicating that the coexisting Cu$^+$ (which acted as a predominant oxidation state)[54] and Cu$^{2+}$ carried a partial positive charge between 0 and +2. Meanwhile, the related FT k$^3$-weighted Extended X-ray absorption fine structure (EXAFS) spectrum in the R space for $Cu_{SA}C_6N_6$ was also measured (Fig. 2c). The presence of a characteristic peak located at 1.51 Å indicated the first coordination shell of Cu-N and there were no obvious metallic Cu−Cu and Cu−O−Cu interaction at around 2.2 and 2.7 Å, respectively[55], demonstrating that Cu atoms were atomically dispersed in the $Cu_{SA}C_6N_6$ matrix. This result was consistent with the HAADF-STEM images. To further explore the atomic coordination of Cu, the

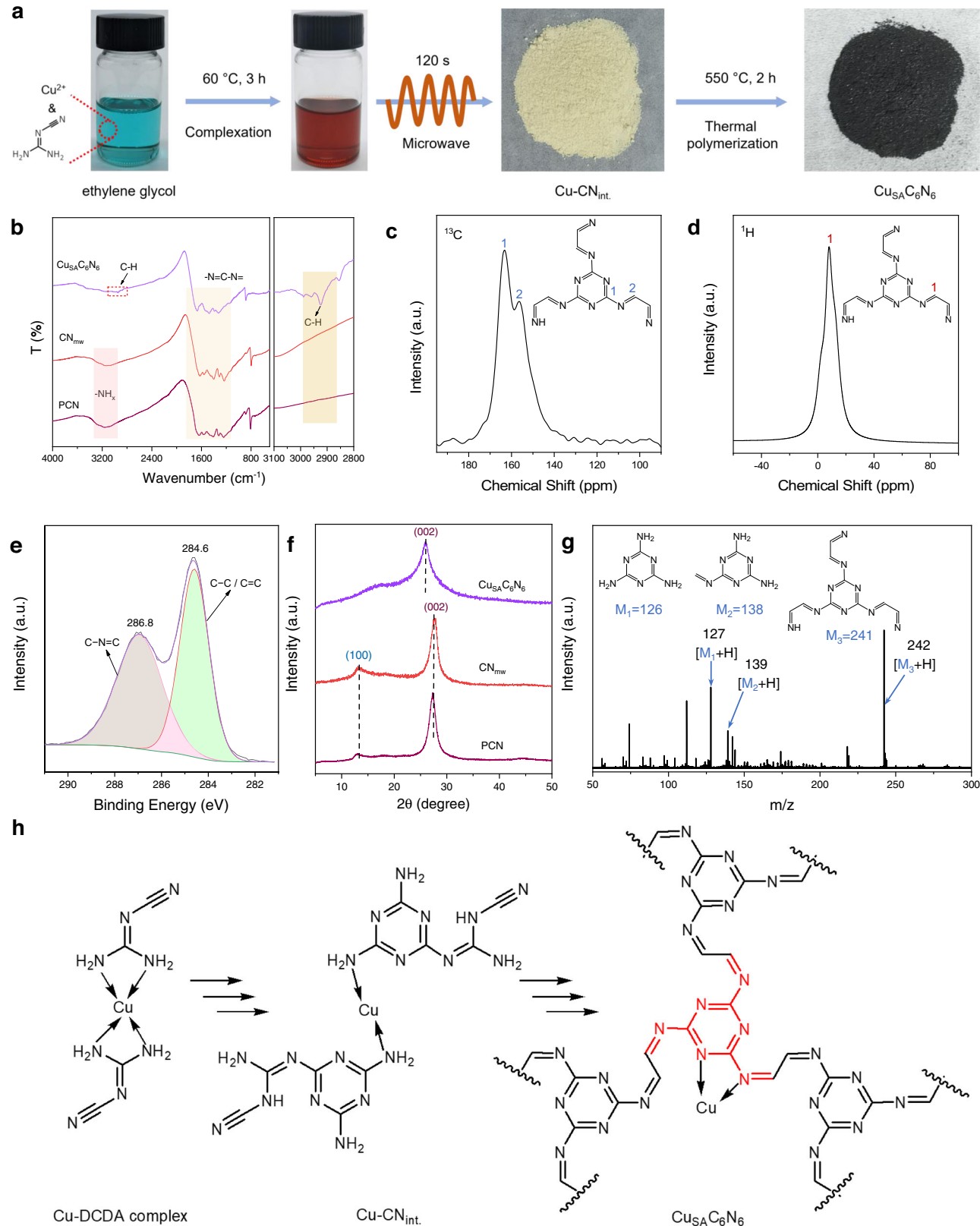

**Fig. 1 | Preparation and molecular structure of Cu$_{SA}$C$_6$N$_6$. a** Brief synthesis procedure for Cu$_{SA}$C$_6$N$_6$. **b** FTIR spectra of Cu$_{SA}$C$_6$N$_6$, CN$_{mw}$ and PCN. Solid-state **c** $^{13}$C and **d** $^1$H NMR spectra of Cu$_{SA}$C$_6$N$_6$. **e** C 1 *s* XPS spectrum of Cu$_{SA}$C$_6$N$_6$. **f** Normalized XRD patterns of Cu$_{SA}$C$_6$N$_6$, CN$_{mw}$, and PCN. **g** LDI-TOF mass spectrum of Cu$_{SA}$C$_6$N$_6$. **h** Proposed condensation processes and molecular structure of Cu$_{SA}$C$_6$N$_6$.

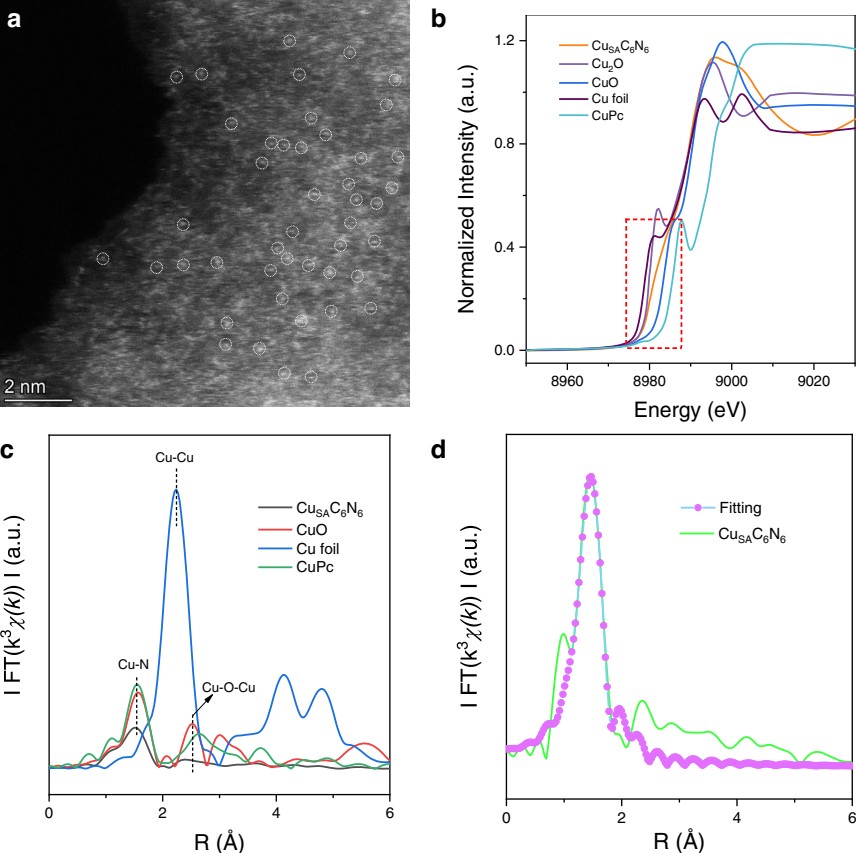

**Fig. 2 | Morphologies and synchrotron XAFS measurements of $Cu_{SA}C_6N_6$.**
**a** HAADF-STEM image of $Cu_{SA}C_6N_6$. **b** Cu k-edge XANES profiles of Cu foil, $Cu_2O$, CuO, $Cu_{SA}C_6N_6$, and CuPc. **c** Cu k-edge EXAFS spectra in the R-space of the $Cu_{SA}C_6N_6$, CuPc, CuO, and Cu foil samples. **d** EXAFS fitting curve for $Cu_{SA}C_6N_6$ in R-space.

related FT $k^3$-weighted EXAFS fitting in R-spaces (Fig. 2d) was performed to reveal the detailed structural information[56]. The fitting analysis result (Supplementary Table 2) showed that the average Cu-N coordination number in the first coordination shell was 2.0 and the Cu-N bond length was 1.93 Å, indicating that one Cu atom was coordinated by two N atoms, forming a $Cu-N_2$ moiety.

The existence of Cu-N chemical bonding information was also verified by the N 1s and Cu 2p XPS spectra. As shown in Supplementary Fig. 9, the N 1s peak could be deconvoluted into four peaks, in which the dominant N species around 398.0 eV was mainly ascribed to the formation of pyridinic nitrogen (C=N-C) in triazine rings; The peak at 399.7 and 400.7 eV were corresponding to pyrrolic N and graphitic N. Interestingly, a new bond at 398.7 eV was observed, demonstrating the formation of a Cu-N bond, which was also in agreement with the EXAFS spectrum in the R-space measurements (Fig. 2c). The Cu 2p spectrum (Supplementary Fig. 10) exhibited two main peaks with a binding energy of 932.9 and 952.9 eV, which were assigned to Cu $2p^{3/2}$ and Cu $2p^{1/2}$, respectively. Furthermore, a weak satellite peak was recorded at 944.8 eV, indicating the presence of $Cu^{2+}$ species in $Cu_{SA}C_6N_6$. The spectrum for Cu $2p^{3/2}$ could be deconvoluted into two peaks at 932.6 and 934.8 eV, corresponding to $Cu^{1+}$ and $Cu^{2+}$, respectively[55,57]. This result further proved the coexistence of $Cu^+$ and $Cu^{2+}$ in $Cu_{SA}C_6N_6$, consisting of the XANES spectroscopy (Fig. 2b).

It should be noted that $Cu_{SA}C_6N_6$ in this work was practically a transition metal complex of a conjugated polymer. But unlike conventional polymers that demonstrate molecular behaviors, polymeric carbon nitrides are almost not dissolvable like graphite; thus, it is often used to support single-atom metals in catalysis[58,59]. The control experiments and comprehensive characterizations, such as XRD, high-resolution TEM, HAADF-STEM, EXAFS, and XPS, collaboratively

demonstrated that Cu emerged as a single-atom state in the $C_6N_6$ matrix, rather than Cu/CuO nanoparticles or clusters. To keep consistency with previous reports[22,32,54,58–61], the term of graphitic $C_6N_6$-based copper single-atom catalyst ($Cu_{SA}C_6N_6$) is used in this study.

## Basic catalytic activity and Gain effect of $Cu_{SA}C_6N_6$

The filling of Cu-N coordination into the $C_6N_6$ framework endowed it with basic peroxidase-like catalytic activity. As shown in Fig. 3a and Supplementary Fig. 11, taking catalytic oxidation of 2, 2′-azino-bis (3-ethylbenzothiazoline-6-sulfonic acid) (ABTS) in the presence of $H_2O_2$ as the model reaction, an evident color change from transparency to green was observed[62]. Moreover, owing to the π-conjugated hybridization of energy levels along the polymer linker and π-stacking between linkers, $Cu_{SA}C_6N_6$ demonstrated a narrow optical gap (1.30 eV, Supplementary Fig. 12 and Fig. 13). Thus, $Cu_{SA}C_6N_6$ was able to effectively utilize the lower excitation energy of light to realize a gain. As shown in Fig. 3b, when irradiated by a household white LED lamp (400–900 nm, 50 mW/cm$^2$, Supplementary Fig. 14), the color of $ABTS_{ox}$ turned dark green, indicating a considerable enhancement in the oxidation of ABTS. The quantitative absorbance of $ABTS_{ox}$ at 417 nm under light irradiation (0.72) was 3.6 times that in the dark (0.20). This result was equivalent to an improved total peroxidase-like rate constant of 3.4 times (Supplementary Fig. 15).

In contrast to $Cu_{SA}C_6N_6$, the basic catalytic activities of control PCN and $CN_{mw}$ were negligible (Fig. 3b). Under light irradiation, their catalytic activities were enhanced, indicative of typical photocatalysts, but were still much lower than that of $Cu_{SA}C_6N_6$. This result demonstrated that the basic activity of photocatalysts without external light irradiation is essentially lacking, which is indispensable for maintaining imperative activity in normal mode. To further understand the

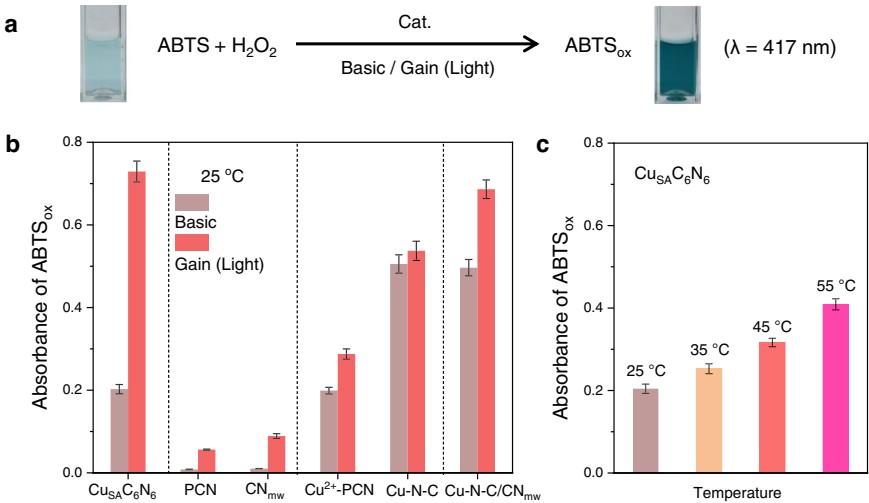

**Fig. 3 | Gain effect evaluation by the standard ABTS catalytic oxidation.**
**a** Equation and photographs of standard ABTS catalytic oxidation using different catalysts as shown in (**b**) and (**c**). **b** Absorbance of $ABTS_{ox}$ catalyzed by $Cu_{SA}C_6N_6$, PCN, $CN_{mw}$, $Cu^{2+}$-PCN, Cu-N-C, and Cu-N-C/$CN_{mw}$ without (basic) and with (gain) light irradiation. **c** Absorbance of $ABTS_{ox}$ catalyzed by $Cu_{SA}C_6N_6$ at different temperatures. ABTS: 2, 2'-azino-bis (3-ethylbenzothiazoline-6-sulfonic acid). Error bars represent the standard error derived from three independent measurements.

excellent basic activity and gain the effect of $Cu_{SA}C_6N_6$, two well-known copper-containing nanozymes, i.e., $Cu^{2+}$-PCN and Cu-N-C with similar Cu-N coordination were also prepared. Interestingly, although higher basic activities of $Cu^{2+}$-PCN and Cu-N-C were observed (Fig. 3b), none of them demonstrated a similar large gain effect as $Cu_{SA}C_6N_6$ under light irradiation, indicating that the second reaction pathway was not effectively initiated. For the former, although charge separation would occur in the $C_3N_4$ framework under light irradiation, the Cu ions significantly quenched it as recombination sites. For the latter, the graphitic carbon framework has intrinsically poor charge-separation ability. These results indicated that conventional carbon nitride-based (photo)catalysts were only applicable for effectively driving a single enzyme-like or photocatalytic reaction because of the negative internal interference or inconsistent reaction conditions in the basic and gain reactions. It was supposed that the unique conjugated linkers (-N=CH-CH=N-) not only elongated the distance of D-A to reduce recombination at Cu sites but also simultaneously compensated for the charge transfer between D-A in $Cu_{SA}C_6N_6$, thus coordinating the basic and gain reactions.

As additional controls, two highly efficient enzyme-like catalysts and photocatalysts, Cu-N-C and $CN_{mw}$, were mixed into a nanocomposite (Cu-N-C/$CN_{mw}$, Supplementary Figs. 16, 17). As shown in Fig. 3b, Cu-N-C/$CN_{mw}$ exhibited a gain effect under light irradiation; however, it reached only approximately one-third of that of $Cu_{SA}C_6N_6$. A series of characterizations, including electrochemical impedance spectra (EIS, Supplementary Fig. 18), photoluminescence (PL) spectra (Supplementary Fig. 19), and photoelectrochemical measurements (Supplementary Fig. 20) of PCN, Cu-N-C/$CN_{mw}$, and $Cu_{SA}C_6N_6$ demonstrated a faster velocity of intramolecular charge migration in the $C_6N_6$ matrix than in PCN and Cu-N-C/$CN_{mw}$ played a crucial role in the boosted gain effect.

Considering that temperature induction is often used to obtain the gain effect, particularly in the photothermal manner in tumor therapies[63], the influence of temperature on ABTS oxidation was also explored. As shown in Fig. 3c and Supplementary Fig. 21, the peroxidase-like activity of $Cu_{SA}C_6N_6$ showed elevated velocity with the increase of temperature. When the temperature was increased from room temperature (25 °C) to 55 °C, the maximum limit for most lives that could endure, the catalytic activity reached 2 times of the original one, but was much smaller than that by mild light irradiation (3.6 times). In this sense, the gain reaction driven by the photocatalytic

method was more efficient by a factor of 80% than that driven by the thermal stimuli.

The intrinsic mechanism of enhancement of the peroxidase-like activity of $Cu_{SA}C_6N_6$ under light irradiation was further investigated. In the first set of experiments, the temperature of the reactor after irradiation using the LED lamp was measured to exclude the photothermal effect. It was found that the irradiation for 10 min made the temperature only improve by 3 °C (Supplementary Fig. 22), whereas the practical reaction in this study only took 3 min, indicating an even smaller temperature fluctuation. The enhancement of peroxidase-like activity with such a minor increase in temperature was further measured (Supplementary Figs. 23, 24), confirming that the photothermal-induced gain effect here was marginal. A series of control experiments, including examination of the oxidase-like activity of $Cu_{SA}C_6N_6$, direct photocatalytic oxidation of ABTS, and decomposition of $H_2O_2$, were also performed, which excluded the potential interferences for the profound gain of peroxidase-like activity (Supplementary Fig. 25). Moreover, the basic catalytic activity and photocatalytic processes of $Cu_{SA}C_6N_6$ can be easily modulated by tuning the Cu content and irradiation power density (Supplementary Figs. 26, 27).

### Coordinated basic and gain reactions mechanism of $Cu_{SA}C_6N_6$
To understand the coordinated basic and gain reaction mechanisms of $Cu_{SA}C_6N_6$ in peroxidase-like activity under light irradiation, the possible intermediate reactive species were first studied using scavenger trapping experiments. As illustrated in Supplementary Fig. 28, both superoxide dismutase (SOD) and isopropanol, corresponding to the superoxide ($O_2^{\cdot-}$) and hydroxyl radical ($\cdot OH$) scavengers, respectively, had negligible influence on the oxidation of ABTS catalyzed by $Cu_{SA}C_6N_6$, indicating that these two radicals were not formed during the activation of $H_2O_2$. Nevertheless, the catalytic activity was notably reduced when isopropanol was added to the reaction solution under light irradiation (Supplementary Fig. 29), indicating that the redox reaction of $H_2O_2$ generated $\cdot OH$ as a major step in $Cu_{SA}C_6N_6$ catalyzed ABTS oxidation. Other trapping experiments using radical probes such as nitrotetrazolium blue chloride (NBT) and coumarin (Supplementary Figs. 30, 31), and electron spin resonance (ESR) spectra (Fig. 4a, b) also supported this speculation. For instance, there was no ESR signal for any ROS-trapping agent adduct, reminiscent of the case catalyzed by HRP. Interestingly, under light irradiation, a typical characteristic peak of the DMPO-·OH spin adduct with a typical signal intensity of 1:2:2:1

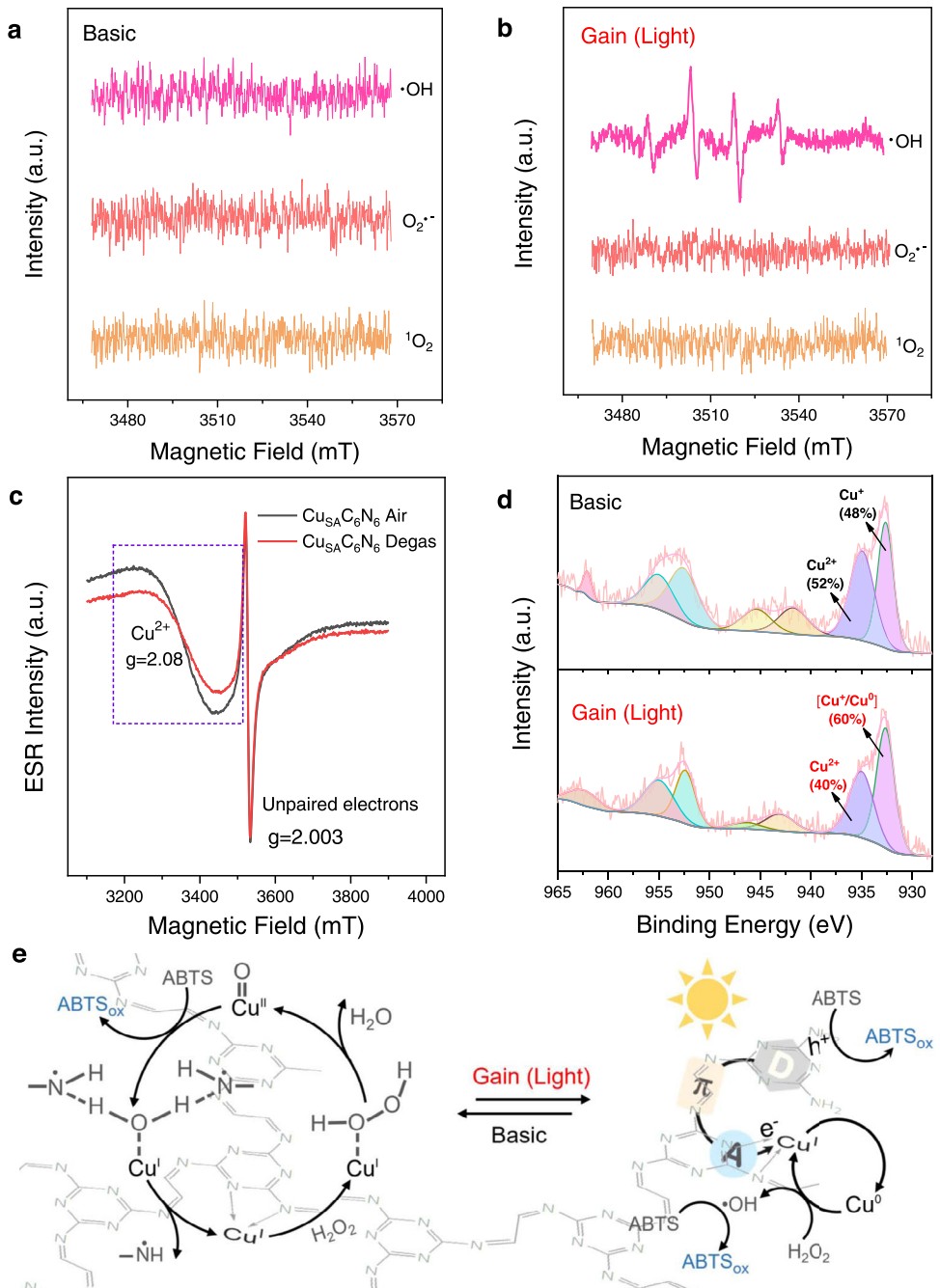

**Fig. 4 | Basic and gain reaction mechanisms for Cu$_{SA}$C$_6$N$_6$.** ESR spectra of the spin adduct of ·OH, O$_2^-$, and $^1$O$_2$ generated during the activation of H$_2$O$_2$ by Cu$_{SA}$C$_6$N$_6$ in 0.2 M HAc·NaAc (pH 5.0) under the **a** basic and **b** gain reactions. **c** EPR spectra of Cu$_{SA}$C$_6$N$_6$ in air and after degassing at 200 °C in vacuum for 12 h. **d** XPS spectra of dynamic changes in the valence state of Cu$_{SA}$C$_6$N$_6$ in the light-on and light-off states. **e** Proposed mechanism for dual peroxidase-like and photocatalytic pathways mimicking the basic activity and gain effect using Cu$_{SA}$C$_6$N$_6$.

was observed (Fig. 4b), indicating the existence of ·OH, in agreement with the scavenger trapping experiments in Supplementary Fig. 28.

The solid-state ESR signal of Cu$_{SA}$C$_6$N$_6$ (Fig. 4c) further demonstrated the probability of the formation of Cu=O species during the peroxidase-like reaction. Given that Cu had a much stronger ESR intensity in the air than in degassed conditions at g = 2.08, it was speculated that Cu$_{SA}$C$_6$N$_6$ might have a similar catalytic mechanism to HRP via a bound ROS pathway[45]. Meanwhile, to investigate the charge transfer as well as the chemical-bond evolution of Cu$_{SA}$C$_6$N$_6$ under light irradiation, the synchronous illumination X-ray photoelectron spectroscopy technique was employed to clarify the dynamic changes of Cu-N[60]. As shown in Fig. 4d, the spectrum for Cu 2$p^{3/2}$ was

deconvoluted into two peaks at 932.6 and 934.8 eV, which were assigned to Cu$^{1+}$ and Cu$^{2+}$, respectively. Interestingly, the percentage of Cu$^{2+}$ decreased from 52 to 40% after light irradiation, while the percentage of Cu$^{1+}$ and Cu$^0$ increased from 48 to 60%, supporting the acceptance of electrons for Cu atoms in the gain reaction[64]. Besides, as a significant amount of Cu$^{2+}$ was retained under light irradiation, the basic reaction pathway should also be activated during the gain reaction, that was much more efficient.

Taking all these experimental evidences into consideration, the mechanism for the light-gained peroxidase-like catalytic activity of Cu$_{SA}$C$_6$N$_6$ was proposed (Fig. 4e). Briefly, in the basic reaction, H$_2$O$_2$ was first bound to the N-coordinated metal site of Cu$^I$ in Cu$_{SA}$C$_6$N$_6$ to

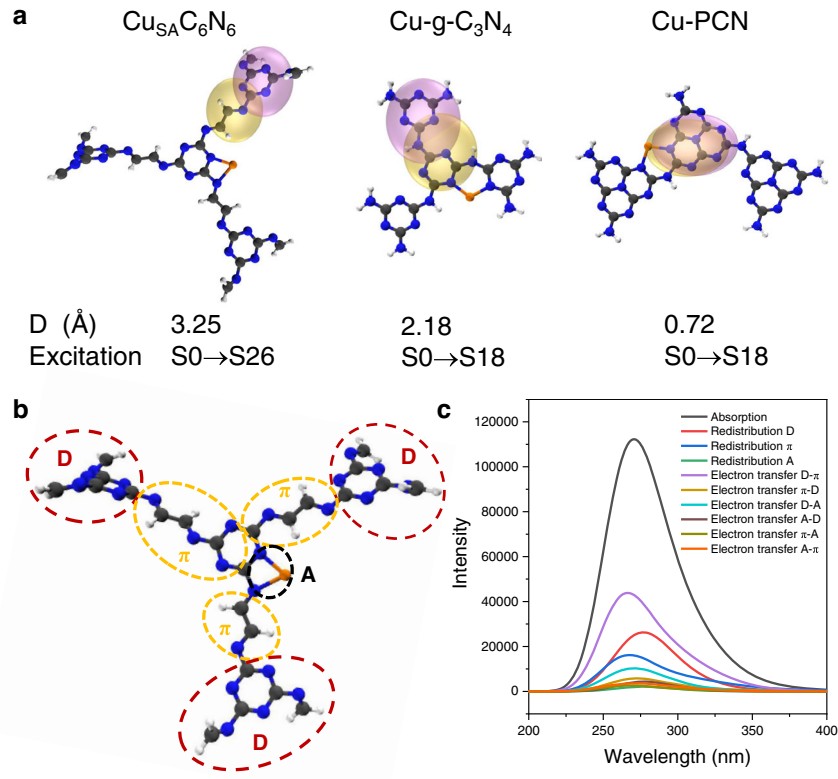

**Fig. 5 | Hole-electron analysis and charge-transfer spectra (CTS) of Cu$_{SA}$C$_6$N$_6$.**
**a** Isosurfaces of hole and electron distribution of the highest intensity of delocalization excitation for Cu$_{SA}$C$_6$N$_6$, Cu-g-C$_3$N$_4$, and Cu-PCN. The corresponding excitation and electron-charge center-of-mass distance (D) were marked below the picture. Calculated smooth description of electron (yellow isosurface) and hole (pink isosurface) spatial population distributions (isovalue = 0.001 au),

respectively. **b** Ball-and-stick model structure of Cu$_{SA}$C$_6$N$_6$. Dashed areas indicate donor (D), π-conjugated charge transfer channels (π), and acceptor (A).
**c** Simulated absorption spectrum and CTS of Cu$_{SA}$C$_6$N$_6$. Electron excitations were calculated with M06-2X/def2-TZVP level based on optimized ground-state geometries. Carbon atoms (black), Nitrogen atoms (blue), Hydrogen atoms (white), Cu atoms (orange).

form Cu$^I$-superoxo species. Then, O-O cleavage occurred to decompose the adsorbed H$_2$O$_2$ into H$_2$O upon the oxidation of Cu$^I$ to Cu$^{II}$=O. Next, the Cu$^{II}$=O was reduced by the ABTS substrate owing to the affinity of ABTS for Cu$^{II}$=O and the electron-donating ability of ABTS. In contrast, when the light was turned on, the photocatalytic processes were activated, providing a new pathway for enhancing the catalytic activity. The terminal mixed-valence Cu species emerged as collectors of photogenerated electrons, and these photoelectrons subsequently reduced Cu$^+$ to Cu$^0$. Subsequently, Cu$^0$ contributed to the in situ decomposition of H$_2$O$_2$ to produce ·OH via a Fenton-like process[61]. Finally, H$_2$O$_2$ oxidized Cu$^0$ to promote the formation of Cu$^+$ species, completing the photocatalytic cycle and maintaining the mixed-valence states. Therefore, for Cu$_{SA}$C$_6$N$_6$, the multiformity of reactive oxygen-related intermediates for the same oxidation reaction made the reaction conditions capable of being the same.

The aforementioned results further demonstrated that Cu$_{SA}$C$_6$N$_6$ not only had a peroxidase-like Cu-N coordination active center but also owned an unusual donor-π-acceptor (D-π-A) unit (Fig. 1h)[65–67], where the single Cu atom acted as an electron acceptor, the triazine rings emerged as electron donors (photovoltaic center), and the -N=CH-CH=N- linkers offered π-conjugated charge transfer channels for D-A couples. Notably, as aforementioned in Fig. 3b, these π-interconnected D-A couples played a crucial role in addressing the negative internal interferences of basic and gain reactions by not only reducing recombination (control sample: Cu$^{2+}$-PCN) and promoting charge separation (control sample: Cu-N-C), but also accelerating intramolecular charge transfer (control sample: Cu-N-C/CN$_{mw}$).

These experimental results were further supported by density functional theory (DFT) calculations. As controls, two more samples were computed. One was Cu-g-C$_3$N$_4$, which was made from Cu$_{SA}$C$_6$N$_6$, but the -N=CH-CH=N- linkers were substituted with N atoms. The other was the most studied Cu-PCN, for which, the triazine ring in Cu-g-C$_3$N$_4$ was further altered into the heptazine ring. The first 50 excited states of these three systems were calculated by using the time-dependent DFT (TD-DFT) method, and the absorption spectra were simulated (Supplementary Fig. 32). The three highest intensity of delocalization excitations were selected for comparison for each molecule. The electron-charge center-of-mass distance (D) of Cu$_{SA}$C$_6$N$_6$, Cu-g-C$_3$N$_4$, and Cu-PCN demonstrate the crucial role of π-interconnected D-A couples (Fig. 5a and Supplementary Fig. 33). The hole-electron analysis was performed using Multiwfn[68,69]. Based on the hole-electron theory, the D value was applied to evaluate the hole-electron separation, and a larger D value was indicative of a more evident hole-electron separation. For Cu-g-C$_3$N$_4$ and Cu-PCN, the D value were 0.24 Å/2.08 Å/2.18 Å and 0.72 Å/0.72 Å/3.08 Å, respectively. Such short electron-charge center-of-mass distance leading to rapid electron-hole recombination without outside assistance. Interestingly, after inserting the -N=CH-CH=N- linkers in Cu$_{SA}$C$_6$N$_6$, the attraction of metal atoms and electrons was reinforced: the electron center had a strong tendency to approach the Cu atom, while the hole center remained concentrated in the excitation triazine ring, resulting in a further increasing the centroid distance of the electrons and holes to 3.08 Å/3.25 Å/3.53 Å. Such sound spatial separation would not only reduce the recombination of electrons and holes, but also promote intramolecular charge separation and migration, which well addressed the negative internal interference

of basic and gain reactions. Furthermore, the most possible D-π-A electron transfer was qualitatively evaluated by analyzing the transitions form from the occupied molecular orbitals to the unoccupied molecular orbitals (Supplementary Figs. 34–36). It was observed that the electron distribution of the occupied molecular orbitals mainly resided on the triazine/heptazine rings in $Cu_{SA}C_6N_6$, Cu-g-$C_3N_4$, and Cu-PCN. While for the unoccupied molecular orbitals, delocalization electrons in $Cu_{SA}C_6N_6$ were transferred from three edge triazine rings to both -N=CH-CH=N- linkers and Cu atom, whereas electrons in Cu-g-$C_3N_4$ and Cu-PCN transitioned to other triazine/heptazine ring, resulting in few electrons delocalized on the Cu atom moieties.

To further understand the inter-fragmental charge transfer during the first 50 excited states in $Cu_{SA}C_6N_6$, the absorption spectra were deconvoluted into charge-transfer spectra (CTS), which were used to visually observe the contribution of each fragment in the model molecules[70]. Dashed areas in the ball-and-stick model structure of $Cu_{SA}C_6N_6$ (Fig. 5b) indicated donor (D), π-conjugated charge transfer channels (π), and acceptor (A). For simplicity, the D-π-A electron transfer was divided into D-π and D-A transitions. As shown in Fig. 5c, the local excitations in $Cu_{SA}C_6N_6$ were mainly concentrated in the D (23.5%) and π (14.6%), and the strongest charge transfer transitions were D-π (38.9%) and D-A (9.6%), in contrast, the electronic transitions from A and π to D were negligible. It was suggested that except for the local transition of the D and π parts, the excitation of the charge separation mainly consisted of D-π and D-A. A series of computational descriptions of the $Cu_{SA}C_6N_6$ systems, including conformational symmetries and the involvement of the transition metal (Supplementary Fig. 37), delocalization (Supplementary Fig. 38), conformational flexibility (Supplementary Fig. 39), and hybrid functionals (Supplementary Fig. 40) were also considered, which supported the above calculation. Therefore, owing to the existence of π-conjugated linkers, $Cu_{SA}C_6N_6$ essentially underwent intramolecular charge transfer from the triazine ring unit to the Cu atom (i.e., D-π-A) upon light irradiation.

## Adaptable sensing of glucose

Improving quality of life is an enduring topic in modern society. Therefore, the intelligent response of glucose has received increasing attention as a powerful tool in the field of human health. In recent years, many biosensing methods for detecting glucose based on nanozymes have found broad utility owing to their simplicity, sensitivity, and high selectivity[71–73]. Despite enormous advances, in vitro recording of the narrow linear concentration range of glucose cannot directly provide physiological information for better comprehension of the dynamic fluctuation of glucose in the brain[74]. An intelligent response to glucose with a random concentration range at a single sensing interface in vivo is highly envisioned to understand the pathological process, but it still faces great challenges.

As proof of this concept, we propose an intelligent response sensor for monitoring glucose in vitro (Fig. 6a). As shown in Fig. 6b, for the conventional biosensors (0 W/cm²), at a low concentration range, the absorbance of ABTS$_{ox}$ at 417 nm was proportional to the first order with glucose. In the high concentration range, the absorbance reached a plateau, leading to a limited detection range. In this sense, relating the response with full-scale analyte concentrations at a logarithmic scale primarily considering a fitting coefficient of determination ($R^2$) close to 1 is widely used, but is questioned in uncertainty in theory[75]. In principle, a reliable fitting should follow the signal transformation mechanism rather than merely the data statistics[76]. Nonetheless, although the strict nonlinear rate equations (Fig. 6b, inset) could match the experimental data well (see the fitted calibration curve in Fig. 6b), it was noted that at the plateau region, the slopes were essentially zero, indicating an extremely poor sensitivity.

Developing adaptive sensitivity and linear detection concentration ranges is an ideal solution to obtain a strict full-scale range of reliable detection. As shown in Fig. 6b, glucose was normally detected

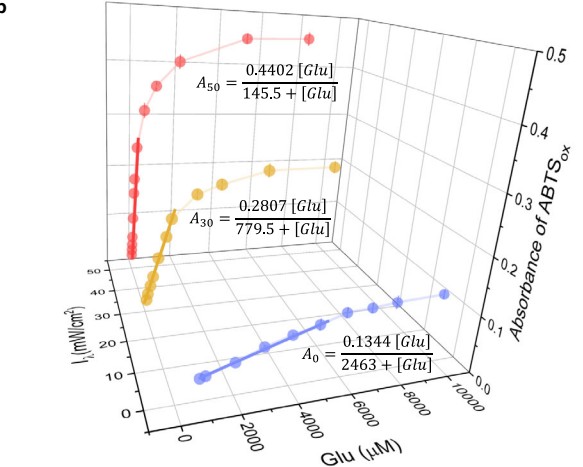

**a**

Basic: $[ABTS_{ox}] = k_B[Glu] + c_1$

Gain: $[ABTS_{ox}] = (k_G + k_B)[Glu] + c_2$ $\quad(I_\lambda\uparrow, k_G\uparrow)$

Glucose
$O_2$

Glucose
acid

GOx

ABTS

$H_2O_2$

$Cu_{SA}C_6N_6$

**b**

$A_{50} = \dfrac{0.4402\,[Glu]}{145.5 + [Glu]}$

$A_{30} = \dfrac{0.2807\,[Glu]}{779.5 + [Glu]}$

$A_0 = \dfrac{0.1344\,[Glu]}{2463 + [Glu]}$

Absorbance of ABTS$_{ox}$

$I_\lambda$ (mW/cm²)

Glu (μM)

**Fig. 6 | Adaptable glucose biosensor. a** Principle of adaptable sensing of glucose based on $Cu_{SA}C_6N_6$. Inset: rate equations of basic and gain reactions under tuned light intensity ($I_\lambda$). $k_B$ and $k_G$ are the intrinsic rate constants of the basic and gain reactions, respectively. $c_1$ and $c_2$ are constants. **b** Absorbance of ABTS$_{ox}$ at 417 nm catalyzed by $Cu_{SA}C_6N_6$ with different concentrations of glucose in the presence of glucose oxidase under tuned light intensity (0, 30, and 50 W/cm²). Inset: nonlinear rate equations of basic and gain reactions with different concentrations of glucose. Bold lines are linear relationship between the absorbance of ABTS$_{ox}$ at 417 nm and concentrations of glucose under tuned light intensity. Error bars represent the standard error derived from three independent measurements.

in a linear range of 800–6000 μM. The linear range can be extended from 50–1000 μM at 30 mW/cm² to 5–80 μM at 50 mW/cm². In the same manner, the limit of detection (LOD) for glucose was tuned from 195.02 μM in the normal mode, to 15.07 and 1.20 μM under irradiation at 30 and 50 mW/cm², respectively. The rate equations in Fig. 6a inset (see the detailed elementary reactions and derivations in Supplementary Information) well supported these facts that the concentration of ABTS$_{ox}$ was in linear with that of glucose, and the slope (i.e., sensing sensitivity) was positively correlated with light intensity. Therefore, a full-scale range of reliable detection could be successfully realized by intelligent switching among different ranges of a single sensor in the measuring progress, enabled by the high adaptability of $Cu_{SA}C_6N_6$ using basic and gain reactions.

It should be noted that although several pioneering papers regarding $C_3N_4$-based biosensors have been reported[40,77], to our knowledge, intelligent biosensors with self-adaptability have been rarely reported so far. Unlike most switchable systems that have no activities in the off-state, $Cu_{SA}C_6N_6$ had a considerable basic activity, reminiscent of the living system in maintaining the necessary activity under normal conditions. Indeed, due to limited room, intelligent artificial devices are preferred to equip biosensors as few as possible and mechanically switching of them would lead to a low operation efficiency. In this sense, one single biosensor with an adaptable linear detection range and sensitivity via an automatic light switch would be

helpful. It is foreseeable that further considering the intrinsic outstanding temporal and spatial resolution of light irradiation, this intelligent biosensor would supply a prospective candidate for dynamic chemical noses for intelligent artificial devices, such as robots, brain-machine interface, and internet-to-things in a high level of integration.

## Discussion

In summary, we proposed a $Cu_{SA}C_6N_6$ single-atom catalyst to address the inconsistency of reaction conditions and negative internal interference for basic and gain reactions in mimicking self-adaptability from nature. $Cu_{SA}C_6N_6$ had single Cu atoms, repetitive triazine cores, and -N=CH-CH=N- D-π-A linkers, which successfully coordinated the peroxidase-like Cu·N coordination center and light responsive center. $Cu_{SA}C_6N_6$ could not only drive the basic ABTS oxidation through bound high-valent copper-oxo pathway, but also undertake a second gain reaction triggered by light via a free hydroxyl radical pathway under the same conditions. The multiformity of reactive oxygen-related intermediates for the same oxidation reaction made the reaction conditions capable of being the same. The comprehensive experiments and DFT calculations further verified the unique topological structure of $Cu_{SA}C_6N_6$ along with the specified D-π-A linker promoted intramolecular charge separation and migration, thus successfully inhibiting the negative interference of electron transfers between peroxidase-like Cu·N coordination center and photoresponsive center. As a result, $Cu_{SA}C_6N_6$ demonstrated a sound basic catalytic oxidation activity and a superb gain up to 3.6 times under household light ($50\,mW/cm^2$). This performance was significantly higher than that of the control systems, including peroxidase-like catalysts, photocatalysts, or their mixtures, and even $Cu_{SA}C_6N_6$ that under thermal stimuli (by a factor of 80% rising from room temperature to 55 °C, the maximum temperature for most lives that can endure). As a proof-of-concept application, $Cu_{SA}C_6N_6$ was successfully applied in a biosensor of glucose, which intelligently switched sensitivity and linear detection range in vitro, simply by tuning the light intensity. It is highly envisioned that an adaptable graphitic $C_6N_6$-based copper single-atom catalyst, along with the further intrinsic temporal and spatial resolution of light, would supply a prospective candidate for adaptable chemical noses for artificial devices, such as robots, brain-machine interface, and internet-to-things in a higher level of integration.

## Data availability

The data supporting the conclusions of this study are present in the paper and the Supplementary Information. The raw data sets used for the presented analysis within the current study are available from the corresponding authors upon request.

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

## Acknowledgements

This work was supported by the National Natural Science Foundation of China (22174014 and 22074015), and the Fundamental Research Funds for the Central Universities. We thank Prof. Yingpu Bi (Lanzhou Institute of Chemical Physics) for offering synchronous illumination X-ray photoelectron spectroscopy techniques.

## Author contributions

Y.Z. and Q.H. conceived and designed the experiments. Q.H. performed the synthesis, characterization, activity evaluation, mechanism studies, and intelligent response sensor of $Cu_{SA}C_6N_6$. H.Y. performed the DFT calculation. Y.F. and W.L. assisted in microwave-assisted condensation. C.Z., Z.W., and X.C. participated in the activity evaluation. All authors contributed to the analysis and discussion of the results. Q.H., H.Y., and Y.Z. co-wrote the manuscript and Y.Z., Z.Z., S.L., and Y.S. revised the manuscript. All authors reviewed the manuscript. Y.Z. supervised the project.

## Competing interests

The authors declare no competing interests.
