## [Peer Review File · Nature Communications]

Adaptable Graphitic C₆N₆-Based Copper Single-Atom Catalyst for Intelligent BiosensingReviewers' Comments:

Reviewer #1:

Remarks to the Author:

Manuscript Title: Self-Adapting Graphitic C₆N₆-Based Copper Single-Atom Catalyst for Intelligent Biosensing

Authors: Qing Hong, Hong Yang, Yanfeng Fang, Wang Li, Caixia Zhu, Zhuang Wang, Sicheng Liang, Xuwen Cao, Zhixin Zhou, Yanfei Shen, Songqin Liu, Yuanjian Zhang

Comments:

In this manuscript, the authors constructed copper modified graphitic carbon nitride catalyst for biosensing applications. This is very interesting paper, especially the concept of the single-atom catalysts and the self-adaptability process but the idea of using modified graphitic carbon nitride in the biosensor is not very new (e.g. doi.org/10.1039/D2RA02315F).

Unfortunately I cannot recommend the publication of this paper in present form in Nature Communications due to the following issues:

1) Based on the methodology the authors used for synthesis of these copper modified carbon nitride composites, it is extremely difficult to obtain copper single atom (only) modified composite samples, especially when oven drying is constantly used during synthesis. There must be some obvious nanoparticles of Cu/CuO exist on the sample surface based on the data (e.g., XPS data) the authors provided.

2) The authors showed very few HR-TEM images with poor qualities and stated in the manuscript (for figure 2a) that "the large-area high-angle annular dark-field scanning transmission electron microscopy (HAADF-STEM) image (Fig. 2a) corroborated the existence of uniformly dispersed single-atom Cu on the C₆N₆ matrix, evident by abundant isolated bright spots highlighted by a red circle". It is actually skeptical if these blurred features been circled on this image are Cu single atoms. Better and clearer HR-TEM images and other characterization data should be collected if the authors want to convince the reader that some or most of the copper in the samples exists in single-atoms form. XAFS data only would be far from enough.

3) In this paper, the composites samples were used for glucose biosensing. Additional explanations/descriptions on why this application is chosen for the copper modified carbon nitride should be added. Some literature comparison (of similar materials in similar applications) would be needed.

4) Novelty is very important. The authors need to emphasize the innovative points (if there is any) in the paper. For example, by using some relatively new conceptual terms. In this context, the paper needs to be improved.

Reviewer #2:

Remarks to the Author:

Zhang and co-workers report a self-adaptive Cu SA C₆N₆ single-atom catalyst that can be applied to a glucose biosensor. The topic is interesting. It might be accepted after the following points are addressed.

1. For CuSAC₆N₆, Cu-g-C₃N₄ and Cu-PCN, does Cu coordinate with N atoms? While In Fig. 5, Cu atoms do not link with N atoms.

2. The author stated that "The aforementioned results further demonstrated that CuSAC₆N₆ not only had a peroxidase-like Cu-N coordination active center but also owned an unusual donor-π-acceptor

(D- π -A) unit (Fig. 1h), where the single Cu atom acted as an electron acceptor, the triazine rings emerged as electron donors (photovoltaic center), and the -N=CH-CH=N- linkers offered π -conjugated charge transfer channels for D-A couples." How to get this point? Actually, it can be drawn from TD-DFT calculations. Even though the absorption spectra were simulated, while the electron transitions were not analyzed in this part, so the D- π -A character of CuSAC6N6 needs to be analyzed.

3. The molecular frontier orbital isosurfaces of the LUMO and HOMO for CuSAC6N6, Cu-g-C3N4, and Cu-PCN are from DFT computations which are based on ground states of studied compounds. From Fig. S31-33, it seems to me that those molecular frontier orbital isosurfaces are from TD-DFT computations, it is unreasonable to explain the electron population of LUMO by using Fig. S31-33 (on page 16).

4. The sentence "As controls, the Cu-g-C3N4 (where the N atom substituted the -N=CH-CH=N- linkers) and Cu-PCN were computed (Fig. S30 and Table S3)." is not clear. What were the properties or parameters computed?

Reviewer #3:

Remarks to the Author:

Qing Hong et al. present a C6N6-based copper catalyst whose catalytic activity can be modulated by light exposure. The activity of the catalyst is shown for ABTS and glucose. The idea is very good (although I cannot comment on how good a 3.6x gain is compared to the state of the art). The paper is well-written but some of the terminology should be cleared up. The work features a large number of experimental methods, which describe many different important aspects (but I am not in a position to judge the quality of the individual experiments). Computations were performed, but these are sub-standard. I could not even find what level of theory was used for these computations. I would not say that these computations actually support the conclusions made.

1. The term "self-adapting" in the title is not clear. Isn't the better term "switchable"? The activity has to be switched externally, right?

2. Also the term single-atom catalyst is not ideal. Essentially what the authors are showing is a transition metal complex of a conjugated polymer. This might be a more natural way to discuss this using existing nomenclature. Is every transition metal complex also a "single atom catalyst"?

3. The authors should make it clear what level of theory was used for the computations. This is to make the work reproducible and to give credit to the developers of the methods used. I could not find any indication of the level of theory. This would mean a "fail" even for an undergraduate lab report.

4. Aside from the overall level of theory, it is also not clear what is plotted in Fig. S31. Cu_SA C6N6 should only have one HOMO and one LUMO.

5. The value of Fig. 5 is not clear. Why is the electron/hole distribution in Fig. 5b so different from HOMO and LUMO in Fig. 5a?

6. Generally speaking it is difficult to provide a reliable computational description of these systems, due to delocalization, approximate symmetries, conformational flexibility, and the involvement of the transition metal. It seems like the authors cherry-picked a few results that support their story. But I don't think there is actually much value in the orbital pictures shown.

7. When the excitation goes into the Cu-atom. What kind of orbital is involved?

Point-by-point response to comments for NCOMMS-22-51358

Reviewer 1:

Q: In this manuscript, the authors constructed copper modified graphitic carbon nitride catalyst for biosensing applications. This is very interesting paper, especially the concept of the single-atom catalysts and the self-adaptability process but the idea of using modified graphitic carbon nitride in the biosensor is not very new (e.g.doi.org/10.1039/D2RA02315F).

A: We appreciate the reviewer's comment that "This is **very interesting** paper, especially the **concept** of the single-atom catalysts and the self-adaptability process". We also appreciate the other valuable suggestions, which enables us to further improve the quality of this work.

As pointed out by this reviewer, the work by Khan et al ("Graphitic carbon nitride and APTES modified advanced electrochemical biosensor for detection of 17 β -estradiol in spiked food samples", *RSC Adv.*, 2022, DOI: 10.1039/D2RA02315F) seems similar to ours in that they use a modified graphitic carbon nitride in the biosensor but this is really a **superficial similarity**. **The chemistry is completely not the same**. In this paper they use a 3-aminopropyltriethoxysilane-modified C₃N₄ to detect 17 β -estradiol through an electrochemical probe. In contrast, in our work, **beyond** traditional C₃N₄, C₆N₆, a new carbon nitride family member, was synthesized to support Cu single-atom into a well-defined catalyst. It successfully mimicked the natural adaptability through **independent basic-** and **gain-**transduction pathways under the same physiological conditions. As a **proof-of-concept** application, self-adapting Cu_SA C₆N₆ has been successfully applied to a **stand-alone** glucose biosensor, which can **intelligently** switch the linear detection range and sensitivity in vitro. By contrast, traditional approaches often require **multiple** biosensors with different detection ranges and sensitivity. To our knowledge, **intelligent biosensors** with self-adaptability have been **rarely** reported so far. Therefore, we believe although several pioneering papers regarding C₃N₄-based biosensors have been reported, our work makes a **substantial advance** expected of a paper published in Nature Communications.

For a better scholarly presentation, "*It should be noted that although several pioneering papers regarding C₃N₄-based biosensors have been reported^{40, 76} to our knowledge, intelligent biosensors with self-adaptability have been rarely reported so far.*" has been added in the revised manuscript (Page 21)

Q1: Based on the methodology the authors used for synthesis of these copper modified carbon nitride composites, it is extremely difficult to obtain copper single atom (only) modified composite samples, especially when oven drying is constantly used during synthesis. There must be some obvious nanoparticles of Cu/CuO exist on the sample surface based on the data (e.g., XPS data) the authors provided.

Q2: The authors showed very few HR-TEM images with poor qualities and stated in the manuscript (for figure 2a) that “the large-area high-angle annular dark-field scanning transmission electron microscopy (HAADF-STEM) image (Fig. 2a) corroborated the existence of uniformly dispersed single-atom Cu on the C₆N₆ matrix, evident by abundant isolated bright spots highlighted by a red circle”. It is actually skeptical if these blurred features been circled on this image are Cu single atoms. Better and clearer HR-TEM images and other characterization data should be collected if the authors want to convince the reader that some or most of the copper in the samples exists in single-atoms form. XAFS data only would be far from enough.

A1/2: Thanks for the very careful concerns on the single atom form of the proposed Cu_{SA}C₆N₆. As this reviewer pointed out, it is true that simply pyrolysis of DCDA/Cu mixture without pre-coordination led to the formation of CuO. Indeed, we adopted one of the acknowledged synthetic strategies for single-atom catalysts preparation (Ref. 32: *J. Am. Chem. Soc.* **2018**, *140*, 16936), including pre-assembly, heating (to remove solvent) and high-temperature thermal polymerization in the Cu_{SA}C₆N₆ preparation (**Fig. 1a**).

Fig. 1a Brief Synthesis procedure for Cu_{SA}C₆N₆.

To support the single atom form, several typical characterizations of the as-prepared Cu_{SA}C₆N₆ have been made, e.g., using XRD, TEM and EXAFS to eliminate potential nanoparticles of Cu/CuO that exist on the sample surface. As seen in **Fig. S1**, the XRD of Cu-CN_{int.} prepared by microwave-assisted condensation of DCDA/Cu mixture not only had a peak at 26.7°, but also showed series of peaks at 35.7°, 38.8° and 48.9°, assigning to CuO; while that by microwave-

assisted condensation of **well pre-coordinated** DCDA-Cu (e.g., for 3 h) not. **It clearly indicated the pre-coordinated Cu-DCDA complex prevented the formation of metal or metallic oxide in Cu-CN_{int}.** As a result, after further thermal condensation at 550 °C, the XRD pattern of the final Cu₅A C₆N₆ in **Fig. 1f** only had a peak at 26.0° (002), corresponding to interlayer stacking reflection (002), and no obvious peaks of Cu/CuO nanoparticles were observed. Consistently, The EXAFS spectra in the R space for Cu₅A C₆N₆ (**Fig. 2c**) displayed a characteristic peak located at 1.51 Å indicated the first coordination shell of Cu-N and there was no obvious metallic Cu-Cu and Cu-O-Cu interaction at around 2.2 and 2.7 Å, respectively (Ref. 55: *J. Am. Chem. Soc.* **2021**, 143, 14530).

Fig. S1 XRD patterns of Cu-CN_{int} prepared from Cu/DCDA with different pre-coordination time (0, 1, 2 and 3 h). [NEW data]

Fig. 1f and 2c XRD pattern of Cu_{SA}C₆N₆ and EXAFS spectra of Cu_{SA}C₆N₆ and other control samples [Updated data]

Moreover, it should be noted that C₆N₆ was prepared at 550 °C and had a much lower graphitization degree than those under a higher pyrolysis temperature (e.g., 700-900 °C) in the literatures. It is well known, in the presence of organic substances, acquiring high quality HAADF-STEM image is not an easy task. To address this problem, the **NEW** HAADF-STEM images, high-resolution TEM images and corresponding STEM-EDS elemental mapping images have been measured again on another independent facility operating by a senior scientist.

The high-resolution TEM images in **Fig. S5** revealed the ultrathin nanosheet-like morphology of Cu_{SA}C₆N₆ and no obvious Cu/CuO nanoparticles existed on the surface of C₆N₆ framework, which agreed well with the XRD results (**Fig. 1f**). The large-area high-angle annular dark-field scanning transmission electron microscopy (HAADF-STEM) images (**Fig. 2a** and **Fig. S6**) corroborated the existence of uniformly dispersed single-atom Cu on the C₆N₆ matrix, evident by isolated bright spots highlighted by a white circle. The corresponding high-resolution STEM-EDS elemental mapping images also showed that the C, N, and Cu species were atomically and homogeneously dispersed across the entire C₆N₆ nanosheet (**Fig. S7** and **Fig. S8**). **Those results jointly demonstrate that Cu emerged as single-atom state atomically dispersed in the Cu_{SA}C₆N₆ matrix rather than Cu/CuO nanoparticles.**

Fig. 2a HAADF-STEM image of Cu₅AC₆N₆ [NEW data]

Fig. S5 High-resolution TEM images of Cu₅AC₆N₆ [Updated]

Fig. S6 HAADF-STEM images of $\text{Cu}_5\text{AC}_6\text{N}_6$ in different areas [NEW data]

Fig. S7 HR-STEM EDS mapping images of $\text{Cu}_S\text{A}_C\text{N}_6$ [NEW data]

Fig. S8 HR-STEM EDS mapping images of $\text{Cu}_S\text{A}_C\text{N}_6$ in other area [NEW data]

Therefore, by following the suggestion, we have obtained stronger evidences of single atom in $\text{Cu}_{\text{SAC}}\text{C}_6\text{N}_6$, and the following revisions have been made:

(1) The XRD patterns of $\text{Cu-CN}_{\text{int}}$ have been updated as the control experiment in SI (**Fig. S1**, Page 9).

(2) The HR-TEM images have been updated as more examples in SI (**Fig. S5**, Page 13).

(3) The **NEW** HAADF-STEM images of higher quality have been updated in the revised manuscript (**Fig. 2a**, Page 7) and SI (**Fig. S6**, Page 14). And, the according discussion “... *abundant* isolated bright spots highlighted by a red circle.” has been revised as “... *abundant isolated bright spots highlighted by a white circle.*” in the revised manuscript. (Page 8)

(4) The **NEW** HR-STEM EDS mapping images have been added into the revised SI (**Fig. S7** and **Fig. S8**, Page 15-16).

(5) The EXAFS spectrum of CuO in the R space has been added in the revised manuscript in **Fig. 2c** (Page 7).

(6) For clarity, the original discussion “The presence of a characteristic peak located at 1.51 Å indicated the first coordination shell of Cu-N. Furthermore, there was no obvious metallic Cu-Cu interaction at around 2.2 Å.” has been revised as “The presence of a characteristic peak located at 1.51 Å indicated the first coordination shell of Cu-N *and there were no obvious metallic Cu-Cu and Cu-O-Cu interaction at around 2.2 and 2.7 Å, respectively.*” in the revised manuscript (**Fig. 2c**, Page 8).

Q3: In this paper, the composites samples were used for glucose biosensing. Additional explanations/descriptions on why this application is chosen for the copper modified carbon nitride should be added. Some literature comparison (of similar materials in similar applications) would be needed.

A4: Good question! In general, the well-known glucose biosensing is used as a proof-of-concept application. The literature comparison and the advance of the glucose biosensor in this work are outlined as follows:

(1) $\text{Cu}_{\text{SA}}\text{C}_6\text{N}_6$ not only has peroxidase-like Cu-N coordination center but also a photo-responsive center, which can intelligently switch the linear detection range and sensitivity in vitro. The self-adapting $\text{Cu}_{\text{SA}}\text{C}_6\text{N}_6$ is able to successfully applied to a **stand-alone** biosensor. By contrast, traditional approaches often require **multiple** biosensors with different detection ranges and sensitivity (Ref. 40: Cho et al. *Nat Commun.* **2019**, *10*, 940; Ref. 71: Wang et al. *Anal. Chem.* **2008**, *80*, 2250).

(2) Re-examination of plotting analytical response against different forms of concentration, relating the response with full-scale analyte concentrations at *logarithmic scale* primarily considering a fitting coefficient of determination (R^2) close to 1 is widely used, but is questioned in uncertainty in theory (Ref. 75: Urban et al. *Anal. Chem.* **2020**, *92*, 10210). In principle, a reliable fitting should follow the *signal transformation mechanism* rather than merely the data statistics. Nonetheless, although the strict non-linear rate equations could match the experimental data well, it was noted that at the plateau region, the slopes were *essentially zero*, indicating an extremely *poor sensitivity* (Ref. 76: Zhang et al. *Anal. Chem.* **2021**, *93*, 11910). Therefore, as shown in this work, developing *adaptive sensitivity and linear detection concentration ranges* is an ideal solution to obtain a strict *full-scale range* of reliable detection.

The above discussions were included in the original manuscript. For a better scholarly presentation, they have been polished with more focus on advances in the revised manuscript (Page 19) and the above control literatures have been cited as Ref. 40 and 71.

Q4: Novelty is very important. The authors need to emphasize the innovative points (if there is any) in the paper. For example, by using some relatively new conceptual terms. In this context, the paper needs to be improved.

A5: Yes, some relatively new conceptual terms should be used for emphasize the innovative points in the paper. For this purpose, in the article title, “Self-Adapting” is used to describe the unique feature of $\text{Cu}_{\text{SA}}\text{C}_6\text{N}_6$ and the associated biosensor application, rather than conventional phrase of “Switchable”. Unlike most responsive systems that are completely inactive in the off state, $\text{Cu}_{\text{SA}}\text{C}_6\text{N}_6$ had a considerable basic activity, reminiscent of the livings in maintaining the necessary activity under normal conditions. For a better scholarly presentation, the above

discussion in highlighting the advance of this paper has been added into the revised manuscript (Page 9).

Reviewer 2:

Q: Zhang and co-workers report a self-adaptive Cu SA C₆N₆ single-atom catalyst that can be applied to a glucose biosensor. The topic is interesting. It might be accepted after the following points are addressed.

A: We thank this reviewer rate the topic interesting and offer us valuable suggestions, which we could make a substantial improvement to this manuscript. The detailed response to the comments is shown as follows.

Q1: For CuSAC₆N₆, Cu-g-C₃N₄ and Cu-PCN, does Cu coordinate with N atoms? While in Fig. 5, Cu atoms do not link with N atoms.

A1: We apologize for the confusion. For Cu_{SA}C₆N₆, Cu-g-C₃N₄ and Cu-PCN, Cu coordinates with N atoms, as the relative position of Cu-N was less than 2.0 Å (1.95 Å for Cu_{SA}-C₆N₆, 1.85 Å for Cu-g-C₃N₄ and 1.83 Å for Cu-PCN). It was our mistake to export the images from VMD software without showing the Cu-N bonds.

Therefore, **Fig. 5a** has been corrected in the revised manuscript as follows:

Fig. 5 (a) Isosurfaces of hole and electron distribution of the highest intensity of delocalization excitation for $\text{Cu}_5\text{A}_6\text{C}_6\text{N}_6$, $\text{Cu-g-C}_3\text{N}_4$, and Cu-PCN . The corresponding excitation and electron-charge center-of-mass distance (D) were marked below the picture. Calculated smooth description of electron (yellow isosurface) and hole (pink isosurface) spatial population distributions (isovalue = 0.001 au), respectively. Carbon atoms (black), Nitrogen atoms (blue), Hydrogen atoms (white), Cu atoms (orange). **[REVISED figures]**

Q2: The author stated that “The aforementioned results further demonstrated that CuSAC_6N_6 not only had a peroxidase-like Cu-N coordination active center but also owned an unusual donor- π -acceptor (D- π -A) unit (Fig. 1h), where the single Cu atom acted as an electron acceptor, the triazine rings emerged as electron donors (photovoltaic center), and the $-\text{N}=\text{CH}-\text{CH}=\text{N}-$ linkers offered π -conjugated charge transfer channels for D-A couples.” How to get this point? Actually, it can be drawn from TD-DFT calculations. Even though the absorption spectra were simulated, while the electron transitions were not analyzed in this part, so the D- π -A character of CuSAC_6N_6 needs to be analyzed.

A2: Thanks for your suggestions. Yes, the D- π -A structure was obtained from TD-DFT calculations. Unfortunately, the lost details of the calculations and insufficient discussion of the results led to a misunderstanding. By following your suggestion, we have not only rewritten the original discussion of electron transitions and absorption spectra, but also demonstrated an advanced analytical method of charge-transfer spectra (CTS) **[Fig. 5b, NEW data]** that provided a more comprehensive analysis of the D- π -A character.

Fig. 5 Partition and CTS of $\text{Cu}_{\text{SA}}\text{C}_6\text{N}_6$. (b) Ball-and-stick model structures of $\text{Cu}_{\text{SA}}\text{C}_6\text{N}_6$. Dashed areas indicate donor (D), π -conjugated charge transfer channels (π), and acceptor (A). (c) Simulated absorption spectrum and charge-transfer spectra of $\text{Cu}_{\text{SA}}\text{C}_6\text{N}_6$. Carbon atoms (black), Nitrogen atoms (blue), Hydrogen atoms (white), Cu atoms (orange). [NEW data]

The original discussion of electron transitions and absorption spectra has been re-written as follows in the revised manuscript (Page 16-17):

These experimental results were further supported by density functional theory (DFT) calculations. As controls, two more samples were computed. One was Cu-g- C_3N_4 , which was made from $\text{Cu}_{\text{SA}}\text{C}_6\text{N}_6$, but the $-\text{N}=\text{CH}-\text{CH}=\text{N}-$ linkers were substituted with N atoms. The other was the most studied Cu-PCN, for which, the triazine ring in Cu-g- C_3N_4 was further altered into the heptazine ring. The first 50 excited states of these three systems were calculated by using time-dependent DFT (TD-DFT) method, and the absorption spectra were simulated (Fig. S32). The three highest intensity of delocalization excitations were selected for comparison for each molecule. The electron-charge center-of-mass distance (D) of $\text{Cu}_{\text{SA}}\text{C}_6\text{N}_6$, Cu-g- C_3N_4 , and Cu-PCN demonstrate the crucial role of π -interconnected D-A couples (Fig. 5a and Fig. S33). The hole-electron analysis was performed using Multiwfn. Based on the hole-electron theory, the D value is applied to evaluate the hole-electron separation, and a larger D value indicates a more evident hole-electron separation. For Cu-g- C_3N_4 and Cu-PCN, the D value were 0.24 Å/ 2.08 Å/ 2.18 Å and 0.72 Å/0.72 Å/3.08 Å, respectively. Such short electron-charge center-of-mass distance leading to rapid electron-hole recombination without outside assistance. Interestingly, after inserting the $-\text{N}=\text{CH}-\text{CH}=\text{N}-$ linkers in $\text{Cu}_{\text{SA}}\text{C}_6\text{N}_6$, the attraction of metal atoms and electrons was reinforced: the electron center had a strong tendency to approach the Cu atom, while the hole center remained concentrated in the excitation triazine ring, resulting in a further increasing the centroid distance of the electrons and holes to 3.08 Å/ 3.25 Å/ 3.53 Å. Such sound spatial separation would not only reduce the recombination of electrons and holes, but also promote intramolecular charge separation and migration, which well addressed the negative internal interference of basic and gain reactions. Furthermore, the most possible D- π -A electron-transfer was qualitatively evaluated by analyzing the transitions from the occupied molecular orbitals to the unoccupied molecular orbitals (Figure S34-S36). It was observed that the electron distribution of the occupied molecular orbitals mainly resided on the triazine/heptazine rings in $\text{Cu}_{\text{SA}}\text{C}_6\text{N}_6$, Cu-g- C_3N_4 and Cu-PCN. While for the unoccupied molecular orbitals, delocalization

electrons in $Cu_{SA}C_6N_6$ were transferred from three edge triazine rings to both $-N=CH-CH=N-$ linkers and Cu atom, whereas electrons in Cu-g- C_3N_4 and Cu-PCN transitioned to other triazine/heptazine ring, resulting in few electrons delocalized on the Cu atom moieties.

To further understand the inter-fragmental charge transfer during the first 50 excited states in $Cu_{SA}C_6N_6$, the absorption spectra were deconvoluted into charge-transfer spectra (CTS), which were used to visually observe the contribution of each fragment in the model molecules. Dashed areas in ball-and-stick model structure of $Cu_{SA}C_6N_6$ (**Figure 5b**) indicated donor (D), π -conjugated charge transfer channels (π), and acceptor (A). For simplicity, the D- π -A electron-transfer was divided into D- π and D-A transitions. As shown in **Figure 5c**, the local excitations in $Cu_{SA}C_6N_6$ were mainly concentrated in the D (23.5%) and π (14.6%), and the strongest charge transfer transitions were D- π (38.9%) and D-A (9.6%), in contrast, the electronic transitions from A and π to D were negligible. It was suggested that except for the local transition of the D and π parts, the excitation of the charge separation mainly consisted of D- π and D-A. A series of computational descriptions of the $Cu_{SA}C_6N_6$ systems, including conformational symmetries and the involvement of the transition metal (**Figure S37**), delocalization (**Figure S38**), conformational flexibility (**Figure S39**), and hybrid functionals (**Figure S40**) were also considered, which supported the above calculation. Therefore, owing to the existence of π -conjugated linkers, $Cu_{SA}C_6N_6$ essentially underwent intramolecular charge transfer from the triazine ring unit to the Cu atom (i.e., D- π -A) upon light irradiation.

Q3: The molecular frontier orbital isosurfaces of the LUMO and HOMO for $Cu_{SA}C_6N_6$, Cu-g- C_3N_4 , and Cu-PCN are from DFT computations which are based on ground states of studied compounds. From Fig. S31-33, it seems to me that those molecular frontier orbital isosurfaces are from TD-DFT computations, it is unreasonable to explain the electron population of LUMO by using Fig. S31-33 (on page 16).

A3: Good Comment! Yes, in TD-DFT computations, Gaussian is used to produce wave functions for the ground state, and using the same method to analyze the electron population of unoccupied molecular orbitals may have systematic errors. However, for larger systems (>50 atoms), as a qualitative analysis, the TD-DFT method can achieve a good balance between accuracy and efficiency. The rationality of this method has been validated by different theoretical

research groups (e.g., Lu et al., *Carbon* **2022**, 187, 78; Zhu et al., *J. Mater. Chem. C* **2015**, 3, 138; Hoffmann et al., *Phys. Chem. Chem. Phys.* **2015**, 17, 11990; Chen et al., *J. Am. Chem. Soc.* **2011**, 133, 12085).

Therefore, for a better scholarly presentation, “*Furthermore, the most possible D- π -A electron-transfer was qualitatively evaluated by analyzing the transitions from the occupied molecular orbitals to the unoccupied molecular orbitals (Figure S34-S36).*” has been added into the revised manuscript (Page 16).

Q4: The sentence “As controls, the Cu-g-C₃N₄ (where the N atom substituted the -N=CH-CH=N- linkers) and Cu-PCN were computed (Fig. S30 and Table S3).” is not clear. What were the properties or parameters computed?

A4: Sorry for the confusion. For a more comprehensive discussion, the above sentence has been revised as “*As controls, two more samples were computed. One was Cu-g-C₃N₄, which was made from Cu_{5A}C₆N₆, but the -N=CH-CH=N- linkers were substituted with N atoms. The other was the most studied Cu-PCN, for which, the triazine ring in Cu-g-C₃N₄ was further altered into the heptazine ring.*” in the revised manuscript (Page 16).

Reviewer 3:

Q: Qing Hong et al. present a C₆N₆-based copper catalyst whose catalytic activity can be modulated by light exposure. The activity of the catalyst is shown for ABTS and glucose. The idea is very good (although I cannot comment on how good a 3.6x gain is compared to the state of the art). The paper is well-written but some of the terminology should be cleared up. The work features a large number of experimental methods, which describe many different important aspects (but I am not in a position to judge the quality of the individual experiments). Computations were performed, but these are sub-standard. I could not even find what level of theory was used for these computations. I would not say that these computations actually support the conclusions made.

A: We thank the reviewer for the comment “The idea is **very** good” and valuable suggestions, which we could make a substantial improvement to this manuscript. Moreover, it also came to our

great attention that we should give a better scholarly presentation to avoid the serious scientific misunderstanding of the theoretical aspect of this manuscript. The detailed response to the comments is shown as follows.

Q1: The term "self-adapting" in the title is not clear. Isn't the better term "switchable"? The activity has to be switched externally, right?.

A1: Very insightful suggestion! Indeed, we have considered many pertinent keywords for the title, like responsive, modulated, switchable and activated. "Self-Adapting" was used in the original manuscript title to describe the unique feature of $\text{Cu}_{\text{SA}}\text{C}_6\text{N}_6$ and the associated biosensor application, rather than conventional phrase of "Switchable". Unlike most responsive systems that are completely inactive in the off state, $\text{Cu}_{\text{SA}}\text{C}_6\text{N}_6$ had a considerable basic activity, reminiscent of the livings in maintaining the necessary activity under normal conditions. But considering the trigger is external, for clarity, "Self-Adapting" in the title has been revised as "Adaptable".

Q2: Also the term single-atom catalyst is not ideal. Essentially what the authors are showing is a transition metal complex of a conjugated polymer. This might be a more natural way to discuss this using existing nomenclature. Is every transition metal complex also a "single atom catalyst"?

A2: Thanks for your very careful concerns on the term single-atom catalyst in this work. It is true that not every transition metal complex is a "single atom catalyst" and $\text{Cu}_{\text{SA}}\text{C}_6\text{N}_6$ in this work was a transition metal complex of a conjugated polymer. But unlike most conventional polymers that demonstrate molecular behaviors, polymeric carbon nitrides are almost not dissolvable like graphite, thus it is often used to support single-atom metals in catalysis (Ref. 58: Li et al., *J. Am. Chem. Soc.* **2018**, 140, 11161; Ref. 59: Li et al., *J. Am. Chem. Soc.* **2018**, 140, 16042). To keep consistence with previous reports, the term of "*Graphitic C_6N_6 -Based Copper Single-Atom Catalyst*" was used in the original manuscript.

Therefore, for a more accurate description, "*It should be noted that $\text{Cu}_{\text{SA}}\text{C}_6\text{N}_6$ in this work was a transition metal complex of a conjugated polymer. But unlike most conventional polymers that demonstrate molecular behaviors, polymeric carbon nitrides are almost not dissolvable like graphite, thus it is often used to support single-atom metals in catalysis (ref.58 and 59). To keep*

consistence with previous reports, the term of graphitic C₆N₆-based copper single-atom catalyst is used in this study.” has been added in the revised manuscript (Page 9).

Q3: The authors should make it clear what level of theory was used for the computations. This is to make the work reproducible and to give credit to the developers of the methods used. I could not find any indication of the level of theory. This would mean a "fail" even for an undergraduate lab report.

A3: We apologize for the confusion. During the collaborative revision process, the computational details of the theoretical calculations in the final version was unfortunately lost. In the revised Supporting Information, such critical information, i.e., *“The ground state geometries of all models were optimized using the 6-31G(d) basis set (the hybrid function is the same as in further calculation) without imaginary frequency.”* and *“Based on the optimized structures, electron excitations were calculated by means of the time-dependent density functional theory method, the first 50 excited states were compared with calculations by using three hybrid functionals (M06-2X (54% Hartree-Fock function), PBE0 (25% Hartree-Fock function) and ω B97XD (22.2% Hartree-Fock function)) in conjunction with def2-TZVP basis set in the gas phase.”* have been added in the "Experimental Details" section (Page S8).

Moreover, to improve the reliability of the computation, the level of TD-DFT has been improved from 6-31G(d) in the original manuscript to def2-TZVP in this revised manuscript, which gives the similar computational results.

Q4: Aside from the overall level of theory, it is also not clear what is plotted in Fig. S31. Cu_SA C₆N₆ should only have one HOMO and one LUMO.

A4: Sorry for the improper description and the associated confusion. The **Fig. S31** in the original Supporting Information aims at showing the most possible D- π -A electron-transfer pathways. By following your suggestion, we have revised the descriptions in the theoretical calculations section by changing "HOMO" and "LUMO" to "occupied molecular orbitals" and "unoccupied molecular orbitals", respectively. The serial number of each molecular orbital have been modified to MO^{H-n}

and MO^{L+n} in **Fig. S34-S36**. For example, **Fig. S34** (corresponding to **Fig. S31** in the original Supporting Information) has been revised and is shown as follows:

Fig. S34 Molecular orbital transitions for the delocalization excitations of the top three intensities in $\text{Cu}_5\text{A}\text{C}_6\text{N}_6$: $\text{S0} \rightarrow \text{S26}$, $\text{S0} \rightarrow \text{S33}$, $\text{S0} \rightarrow \text{S36}$. The corresponding oscillator strengths (f) was marked after the excitation. Isovalues (isovalue = 0.02 au) of the HOMO- n (left, $n \geq 0$) and LUMO+ n (right, $n \geq 0$), green and blue regions denote the positive and negative orbital phases, respectively. The numbers in the middle of the arrow denote the contributions of the transition to the corresponding excitation. Carbon atoms (black), Nitrogen atoms (blue), Hydrogen atoms (white), Cu atoms (orange). Electron excitations were calculated with M06-2X/def2-TZVP level based on optimized ground-state geometries. **[Updated data]**

Q5: The value of Fig. 5 is not clear. Why is the electron/hole distribution in Fig. 5b so different from HOMO and LUMO in Fig. 5a?

A5: In **Fig. 5**, the *D* value is the distance between the centroid of the hole and the electron, and a larger *D* value is indicative of a more evident hole-electron separation.

In the original manuscript, the electron/hole distribution in **Fig. 5b** seems different from HOMO and LUMO in **Fig. 5a**. It is because when only one pair of MO transitions is involved in electron excitation, the probability densities of the HOMO and LUMO can be simply described as the hole and electron distributions. For complex molecules, the calculation of the hole and electron distributions involve a linear combination of several pairs of MO transitions, and if the contribution of HOMO to LUMO transition is low, the hole and electron distributions do not exactly correspond to the equivalent surface of HOMO and LUMO. For our work, the differences between electron/hole distribution and LUMO/HOMO in **Fig. 5a** and **Fig. 5b** is ascribed to the second situation.

For clarity, “*The hole-electron analysis was performed using Multiwfn. Based on the hole-electron theory, the *D* value is applied to evaluate the hole-electron separation, and a larger *D* value is indicative of a more evident hole-electron separation.*” has been added into the revised manuscript (Page 16).

Moreover, for a better scholarly presentation, we have added more comprehensive images (**Fig. S34-S36**) describing several highest contribution pairs of MO transitions in the revised Supporting Information. As the original **Fig. 5a** is included in **Fig. S34-S36**, it has been deleted in the revised manuscript.

Q6: Generally speaking it is difficult to provide a reliable computational description of these systems, due to delocalization, approximate symmetries, conformational flexibility, and the involvement of the transition metal. It seems like the authors cherry-picked a few results that support their story. But I don't think there is actually much value in the orbital pictures shown.

A6: Thanks for your valuable suggestions. By following your suggestion, we have considered these critical influences on the computational results in the revised manuscript by using a more

advanced analytical method of charge-transfer spectra (CTS, Ref. 70: Lu et al., *Carbon* **2022**, 187, 78).

Fig. 5 Partition and CTS of $\text{Cu}_S\text{A C}_6\text{N}_6$. (b) Ball-and-stick model structures of $\text{Cu}_S\text{A C}_6\text{N}_6$. Dashed areas indicate donor (D), π -conjugated charge transfer channels (π), and acceptor (A). (c) Simulated absorption spectrum and CTS of $\text{Cu}_S\text{A C}_6\text{N}_6$. Carbon atoms (black), Nitrogen atoms (blue), Hydrogen atoms (white), Cu atoms (orange). [NEW data]

Fig. S37 Partition and CTS of C_6N_6 . (a) Ball-and-stick model structures of C_6N_6 . Dashed areas indicate donor (D), π -conjugated charge transfer channels (π), and acceptor (A). Carbon atoms (black), Nitrogen atoms (blue), Hydrogen atoms (white), Cu atoms (orange). (b) Simulated absorption spectrum and CTS of C_6N_6 . Electron excitations were calculated with M06-2X/def2-TZVP level based on optimized ground-state geometries. [New data]

Involvement of the transition metal and approximate symmetries: For a better scholarly presentation, the structure of $\text{Cu}_{\text{SA}}\text{C}_6\text{N}_6$ and the control C_6N_6 have been additionally calculated. As shown in **Fig. 5b** and **Fig. S37a**, owing to the acceptor was changed from a $=\text{N}-\text{Cu}-\text{N}=\text{}$ to two N atoms, the contribution of D-A in the CTS (**Fig. 5c** and **Fig. S37b**) decreased from 9.6% ($\text{Cu}_{\text{SA}}\text{C}_6\text{N}_6$) to 2.2% (C_6N_6). It was worth noting that the existence of Cu atom in $\text{Cu}_{\text{SA}}\text{C}_6\text{N}_6$ broke the symmetry, demonstrating that $\text{Cu}_{\text{SA}}\text{C}_6\text{N}_6$ accepted electrons more readily than C_6N_6 that were excited from three triazine rings (donor).

Fig. S38 Partition and CTS of $\text{Cu}_{\text{SA}}\text{C}_6\text{N}_6$ with different metal-anchoring positions. (a, c, e) Ball-and-stick model structures of $\text{Cu}_{\text{SA}}\text{C}_6\text{N}_6$ with three possible metal-anchoring positions. Dashed areas indicate donor (D), π -conjugated charge transfer channels (π), and acceptor (A). Carbon atoms (black), Nitrogen atoms (blue), Hydrogen atoms (white), Cu atoms (orange). (b, d, f) Simulated absorption spectrum and CTS of $\text{Cu}_{\text{SA}}\text{C}_6\text{N}_6$. Electron excitations were calculated with M06-2X/def2-TZVP level based on optimized ground-state geometries. [New data]

Delocalization. To comprehensive analysis of the computational description for delocalization, the CTS of $\text{Cu}_5\text{A}_2\text{C}_6\text{N}_6$ with four possible metal-anchoring positions were calculated. In these structures, Cu was anchored to two N atoms at the central/edge triazine ring. As shown in **Fig. 5b**, **S38b**, **S38d**, and **S38f**, the contribution of D-A transfer transition in the CTS were 9.6%, 3.8%, 6.3%, and 7.6%, respectively, much higher than that without Cu anchored (**Fig. S37b**, 2.2%), indicative of an increased delocalization by the Cu atom.

Fig. S39 Effects of conformational flexibility. Energy curve of optimization and corresponding CTS of $\text{Cu}_5\text{A}_2\text{C}_6\text{N}_6$ structures. (a) Energy curve of $\text{Cu}_5\text{A}_2\text{C}_6\text{N}_6$ optimized from the initial guess to the point of minimal energy. The five points that are marked in red correspond to the structure of the calculated CTS shown in (b-f) from left to right, in that order. Electron excitations were calculated with M06-2X/def2-TZVP level based on optimized ground-state geometries. [New data]

Conformational flexibility. To comprehensive analysis of the computational description for conformational flexibility, five typical structures were randomly extracted from the optimization process (**Fig. S39a**). The energy difference between these structures and the energy minimum structure was less than 18 kJ/mol, corresponding to the requirement of rotation energy for the C-C bond in ethane. It was aimed to simulate the different structures arisen from the thermal motion of the molecule. As shown in the CTS of $\text{Cu}_5\text{A}_2\text{C}_6\text{N}_6$ (**Fig. S39b-S39f**), the contribution of D-A

transfer transition ranged from 7.1 to 9.0%, close to that of the energy minimum structure (**Fig 5c**, 9.6%). These results indicated that conformational flexibility almost had no side effect on the excited state via the D- π -A electron-transfer.

Therefore, for providing more reliable computational descriptions of the $\text{Cu}_{\text{SA}}\text{C}_6\text{N}_6$ systems, the suggested consideration of the involvement of the transition metal and conformational symmetries [**Fig. S37, NEW data**], delocalization [**Fig. S38, NEW data**], conformational flexibility [**Fig. S39, NEW data**], and along with the necessary discussion have been added into the revised Supporting Information. Accordingly, “A series of computational descriptions of the $\text{Cu}_{\text{SA}}\text{C}_6\text{N}_6$ systems, including conformational symmetries and the involvement of the transition metal (**Figure S37**), delocalization (**Figure S38**), conformational flexibility (**Figure S39**), and hybrid functionals (**Figure S40**) were also considered, which supported the above calculation. Therefore, owing to the existence of π -conjugated linkers, $\text{Cu}_{\text{SA}}\text{C}_6\text{N}_6$ essentially underwent intramolecular charge transfer from the triazine ring unit to the Cu atom (i.e., D- π -A) upon light irradiation.” has been added into the revised manuscript (Page 17).

Q7: When the excitation goes into the Cu-atom. What kind of orbital is involved?

A7: As discussed in **Q6/A6**, the effective excitation goes into the Cu atom was a delocalization excitation. According to molecular orbital transitions (**Fig. S34**), a representative excitation that goes into the Cu-atom is shown in **Fig. R1**. It mainly involves the transition from the bonding n orbital of the triazine ring to the anti-bonding n orbital of the Cu atom.

Fig. R1 One electron delocalization transition of $\text{S0} \rightarrow \text{S33}$. Isosurfaces (isovalue = 0.02 au) of the HOMO-8 and LUMO+2 of $\text{Cu}_{\text{SA}}\text{C}_6\text{N}_6$, green and blue regions denote the positive and negative orbital phases, respectively. Carbon atoms (black), Nitrogen atoms (blue), Hydrogen atoms (white), Cu atoms (orange). Hydrogen atoms (white), Cu atoms (orange). Electron excitations were calculated with M06-2X/def2-TZVP level based on optimized ground-state geometries. [**For review only**]

Reviewers' Comments:

Reviewer #1:

Remarks to the Author:

I analyzed the responses to my comments. Most of them seem grounded and accurate. However, I am still not fully convinced that these materials are graphitic C₆N₆-based Cu single-atom catalyst or rather graphitic C₆N₆-based very small Cu cluster catalyst.

Reviewer #2:

Remarks to the Author:

I think the authors have addressed the issues from my concerns.
I am satisfied with the revisions.

Reviewer #3:

Remarks to the Author:

The authors have replied to all reviewer comments in great detail. This is well-performed collaborative work. All doubts have been cleared, as far as I can tell.

Point-by-point response to comments for NCOMMS-22-51358A-Z

Reviewer 1:

Q: I analyzed the responses to my comments. Most of them seem grounded and accurate. However, I am still not fully convinced that these materials are graphitic C₆N₆-based Cu single-atom catalyst or rather graphitic C₆N₆-based very small Cu cluster catalyst.

A: We appreciate this valuable suggestion, which enables us to further improve the quality of this work. The control experiments and the additional data of comprehensive characterizations have been supplied in the last revision, but unfortunately, the conclusive discussion was still lacking. For clarity, this additional discussion has been incorporated into the revised manuscript as follows:

“It should be noted that Cu_{SA}C₆N₆ in this work was a transition metal complex of a conjugated polymer. But unlike conventional polymers that demonstrate molecular behaviors, polymeric carbon nitrides are almost not dissolvable like graphite, thus it is often used to support single-atom metals in catalysis.^{58, 59} To keep consistence with previous reports, the term of graphitic C₆N₆-based copper single-atom catalyst is used in this study.”

has been revised as

“It should be noted that Cu_{SA}C₆N₆ in this work was practically a transition metal complex of a conjugated polymer. But unlike conventional polymers that demonstrate molecular behaviors, polymeric carbon nitrides are almost not dissolvable like graphite, thus it is often used to support single-atom metals in catalysis.^{58, 59} The control experiments and comprehensive characterizations, such as XRD, high-resolution TEM, HAADF-STEM, EXAFS and XPS collaboratively demonstrated that Cu emerged as single-atom state in the C₆N₆ matrix, rather than Cu/CuO nanoparticles or clusters. To keep consistence with previous reports,^{22, 32, 54, 58-61} the term of graphitic C₆N₆-based copper single-atom catalyst (Cu_{SA}C₆N₆) is used in this study.” in the revised manuscript (Page 9).

Reviewer 2:

Q: I think the authors have addressed the issues from my concerns. I am satisfied with the revisions.

A: The generous and positive comments from the reviewer are deeply appreciated.

Reviewer 3:

Q: The authors have replied to all reviewer comments in great detail. This is well-performed collaborative work. All doubts have been cleared, as far as I can tell.

A: The generous and positive comments from the reviewer are deeply appreciated.